# Mitigation of Dietary Microplastic Accumulation and Oxidative Stress Response in Rainbow Trout (*Oncorhynchus mykiss*) Fry Through Dietary Supplementation of a Natural Microencapsulated Antioxidant

**DOI:** 10.3390/ani15071020

**Published:** 2025-04-01

**Authors:** İdris Şener, Matteo Zarantoniello, Nico Cattaneo, Federico Conti, Luca Succi, Giulia Chemello, Elena Antonia Belfiore, Ike Olivotto

**Affiliations:** Department of Life and Environmental Sciences, Università Politecnica delle Marche, 60131 Ancona, Italy; idris_943@hotmail.com (İ.Ş.); n.cattaneo@pm.univpm.it (N.C.); f.conti@pm.univpm.it (F.C.); s1114212@studenti.univpm.it (L.S.); g.chemello@staff.univpm.it (G.C.); e.a.belfiore@pm.univpm.it (E.A.B.)

**Keywords:** aquaculture, fish welfare, dietary microplastics, aquafeed

## Abstract

Microplastic (MP) contamination is a globally recognized environmental concern, with the potential to infiltrate aquaculture systems and accumulate in farmed aquatic organisms, posing potential risks to consumers. In this study, the efficacy of a natural microencapsulated antioxidant as a dietary supplement was assessed to mitigate the adverse effects of dietary MP exposure in rainbow trout (*Oncorhynchus mykiss*) fry. Our findings demonstrated that this innovative formulation effectively alleviated MP-induced oxidative stress, reduced intestinal absorption of MPs, and limited their translocation to other tissues. These results provide valuable insights for the aquaculture industry, highlighting a potential strategy to enhance fish welfare and improve the overall quality of aquaculture products.

## 1. Introduction

Microplastics (MPs) are widely detected in aquatic ecosystems, impacting organisms across all trophic levels [1,2]. Besides posing concerns on wild organisms, these contaminants can threaten the food productive sectors, including aquaculture, causing potential risks for the final consumers [3,4]. In both freshwater and marine aquaculture systems, fish are exposed to MPs through multiple pathways, including those suspended in the water column, the degradation of plastic-based farming equipment, and the ingestion of feed inherently contaminated with MPs originating from raw ingredients and packaging materials [5,6,7]. In fish, the adverse effects of MP exposure are influenced by factors such as particle size, concentration, shape, exposure duration, and polymer composition [8,9], with these effects varying depending on the life stage and feeding habits of the farmed species [10,11,12,13]. The ingestion of dietary MP particles larger than 100 µm can cause mechanical damage to the intestinal lining, leading to inflammatory responses [14], whilst smaller MPs up to 20 µm are known to induce an increase in the goblet cells relative abundance without, however, being absorbed at the intestinal level [15,16]. In contrast, MPs smaller than 20 µm can be internalized by enterocytes and subsequently translocated to other organs, where they may accumulate and persist [17,18,19]. It has been demonstrated that the liver serves as the primary organ for MP accumulation, as these contaminants are transported through the bloodstream and retained in hepatic tissue, thereby reducing their distribution to other organs [16,20]. In fish, the MP accumulation in the liver triggers physiological and molecular responses, with oxidative stress being a key consequence [21,22]. Studies on zebrafish (*Danio rerio*) [15,16] and European seabass (*Dicentrarchs labrax*) [23] exposed to dietary MPs have demonstrated an upregulation of antioxidant enzymes, including superoxide dismutase (SOD) and catalase (CAT). These enzymes play a crucial role in mitigating oxidative damage by neutralizing reactive oxygen species and preventing cellular damage [24]. In this context, the dietary inclusion of natural antioxidant molecules could enhance the fish’s intrinsic defense mechanisms, helping to mitigate MP-induced oxidative stress and reduce its detrimental effects on cellular integrity and organ function [23,25,26]. Among the natural molecules with antioxidant properties, the carotenoid astaxanthin (ASX) has been extensively studied for its potential benefits in the aquaculture industry [27]. Particularly, studies have demonstrated other positive effects of ASX supplementation, such as enhanced intestinal development, improved digestion and nutrient absorption, increased antioxidant enzyme activity, and improved growth performance in rainbow trout (*Oncorhynchus mykiss*) [28,29,30]. Natural astaxanthin is significantly more potent than its synthetic counterpart and is primarily extracted from microalgae, such as *Haematococcus pluvialis*, which produces this carotenoid in high concentrations, especially under stress conditions such as UV light exposure or nutrient deprivation [31,32,33,34]. The cultivation of microalgae not only supports the production of high-value compounds like natural astaxanthin but also aligns with environmental sustainability goals by mitigating the impacts of climate change through carbon capture [35]. However, natural ASX, due to its chemical structure characterized by unsaturated hydrocarbons with electron-rich conjugated double bonds, is highly susceptible to degradation by light, heat, and oxygen [36,37]. To enhance the dietary bioavailability of ASX, its stability can be improved through microencapsulation techniques, which protect the carotenoid from environmental factors and improve its delivery via feed to the fish [23,38,39]. The composition of the microencapsulation wall is critical for enhancing the ASX stability [40]. Various materials, such as chitosan, alginate, pectin, maltodextrin, and starch, have been evaluated for their ability to reduce ASX degradation and improve its bioavailability [41,42,43]. Starch is particularly preferred in microencapsulation due to its advantageous properties, including its edibility, natural origin, ease of modification, and biodegradability [44]. Furthermore, the inclusion of starch in the microcapsule’s wall has significant relevance in addressing MP contamination, as it can promote the coagulation of MPs both suspended in water [45,46] and inside the fish gut [16,23]. Particularly, starch-mediated coagulation of MPs in the fish gut presents a promising strategy for reducing their absorption and potential toxic effects, especially in aquaculture. By promoting the aggregation of MPs, starch-based microcapsules can help minimize the risks associated with MP exposure in farmed fish, while also enhancing the stability and bioavailability of bioactive compounds like ASX [16,23]. However, the effectiveness of this coagulation process is influenced by several factors, such as water conditions (e.g., freshwater vs. saltwater), dietary formulations, exposure duration, and the specific gut environment of teleost fish (e.g., the presence or absence of a stomach and the role of acidic digestion). Recent studies have highlighted these variables in species like European seabass (*Dicentrarchus labrax*) and zebrafish (*Danio rerio*) that were fed diets containing fluorescent MP microbeads, demonstrating how these factors influence the effectiveness of MP starch-mediated coagulation [16,23]. Particularly, it has been demonstrated that this process was successful in European seabass (a saltwater fish with stomach) exposed to a dietary MP concentration of 50 mg/kg which, in turn, was not sufficient to trigger coagulation in the stomach-less zebrafish (coagulation only occurred when fish were exposed to a 10× higher MP dietary concentration). Consequently, the effectiveness of this technology should thus be further investigated considering water conditions (freshwater or salt water) and differences in the gut environment of teleost species. For that reason, the present study aimed to evaluate the efficacy of a natural microencapsulated antioxidant in mitigating the adverse effects of dietary MPs (included in the diet at a 50 mg/kg concentration) in rainbow trout fry (as a monogastric carnivorous freshwater fish) over a 60-day feeding trial. Rainbow trout are the second most important species in the global aquaculture scenario (with an overall production exceeding one million tons in 2022), and their farming is commonly characterized by the use of ASX for enhancing their fillet coloration [47,48]. Biometric, histological, and molecular analyses, along with the quantification of MPs in target organs, were used to assess the role of microencapsulated ASX in reducing MP accumulation and oxidative stress in this important aquaculture species.

## 2. Materials and Methods

### 2.1. Ethics

All procedures conducted on fish were approved by the Ethics Committee of the Marche Polytechnic University (Ancona, Italy, n.3—24 November 2022) and the Italian Ministry of Health (Aut. n. 391/2023-PR) and were in accordance with the Italian legislation on experimental animals. To minimize the suffering, animals were anaesthetized prior to sampling (MS222; Merck KGaA, Darmstadt, Germany).

### 2.2. MP Features

Amino formaldehyde polymer (FMv-1.3) fluorescent MP microbeads, characterized by a 1–5 µm range size, emission peak of 636 nm, and excitation peak of 584 nm, were acquired from Cospheric LCC (Goleta, CA, USA; data sheet available at https://www.cospheric.com/fluorescent_violet_tracer_microparticles_2um.htm; accessed on 20 January 2023) [23].

### 2.3. Microencapsulated ASX

The microencapsulation of natural ASX (AstaReal^®^ L10, Nacka, Sweden) was carried out by STM Aquatrade S.r.l. in Ancona, Italy. The microcapsule wall consisted of a matrix primarily composed of gum Arabic (55%) and starch (22%), with additional components including cellulose, sodium ascorbate, and vitamin E. The detailed process of microcapsule preparation is protected under intellectual property rights. The encapsulation was designed to protect the astaxanthin while ensuring its efficient delivery to the feed. Due to the chemical properties of the microcapsules, their contents were released approximately 90 s after immersion in water. Each gram of ASX-containing microcapsules provided a direct delivery of 25 parts per million (ppm) of ASX to the feed.

### 2.4. In Vitro MP Coagulation

In vitro coagulation experiments were performed according to the method determined by Zarantoniello et al. [23]. Briefly, a stock MP solution was prepared by adding fluorescent microbeads to freshwater at 50 mg/L. The stock solution pH was set to 3.7 using 0.1 M HCl to mimic the rainbow trout gastric conditions [49] and was then aliquoted. Each aliquot was added with ASX microcapsules, ASX alone, empty microcapsules, Arabic gum, or starch. The additions were made to match the concentrations used in the feed, accounting for the relative microcapsule composition. To mimic stomach movements and gastric emptying, 1 mL (in triplicate; *n* = 3) from each aliquot was placed in a 35 mm glass bottom dish (GmbH, Gräfelfing, Germany) and observed before and after a 6 h period on a shaker plate (SO3 Orbital Shaker; Stuart Scientific Co., Ltd., Redhill, UK) with a confocal microscope (Nikon A1R, Nikon Corporation, Tokyo, Japan). The eventual occurrence of coagulation events was verified, and coagula, if present, were measured through the NIS-Element software (version 5.21.00, Nikon). After this first step, in all the 35 mm glass bottom dishes, pH was increased to 7.7 by adding NaOH 1 M to reach the conditions of the rainbow trout anterior intestine [50]. The solutions, after a 4 h-period on the SO3 Orbital Shaker, were again examined with a Nikon A1R confocal microscope (Nikon Corporation) for coagulation events detection.

Since this experimental setup was conducted using the same fluorescent MPs microbeads and the same ASX microcapsules (and their relative components) used in Zarantoniello et al. [23], the results obtained in the present study were comparable. Particularly, MP coagulated in the solutions at pH 3.7, containing ASX microcapsules, empty microcapsules, and starch, whilst no coagula were found in those containing MPs alone, ASX alone, and Arabic gum. The same results were obtained after the pH transition from 3.7 to 7.7. Images obtained with the NIS-Element software were similar to those reported in Zarantoniello et al. [23]. For those reasons, only the methodology and the overall results of the in vitro test were reported for the present study.

### 2.5. Experimental Diets

Two distinct control diets were prepared using the same base ingredients. The first control diet, designated as CTRL, was free of fluorescent MPs and ASX and was formulated to mimic the proximate composition of a commercial standard diet for rainbow trout fry, according to Gesto et al. [51]. The second control diet, labeled CTRL-ASX, was prepared by incorporating 7 g of microencapsulated ASX per kg of feed, providing a concentration of 175 mg of ASX per kg of feed. For the treatment groups, two experimental diets were formulated: (i) A50 diet, which included 50 mg of fluorescent MPs per kg of feed added to the CTRL diet formulation (as described in Cattaneo et al. [15]); (ii) A50-ASX diet, which contained both 50 mg of fluorescent MPs and 7 g of microencapsulated ASX per kg of feed.

Prior to the inclusion of ASX microcapsules, diets were weighed and transferred to airtight glass jars, where they were vigorously mixed to ensure uniform distribution of ASX microcapsules across each pellet. ASX content in the diets was determined from three subsamples per diet following the method of Du et al. [52], confirming an average ASX concentration of 172 ± 6 mg/kg feed.

All powdered ingredients for the experimental diets were thoroughly blended (GastroNorm 30C1PN, ItaliaGroup Corporate Srl, Ponte nelle Alpi, Italy) for 20 min. Subsequently, oil and water were added to achieve the required consistency for pelleting. Pellets were then produced using a 3 mm-die meat grinder, dried at 37 °C for 48 h in a ventilated oven, and ground before being sieved to obtain appropriately sized particles for rainbow trout fry (0.6–0.8 mm) [51].

### 2.6. Experimental Design

Rainbow trout eggs at eyed stage from the Eredi Rossi (Rieti, Italy) farm were stored in a 400 L tank with an open water circuit, maintained at 16.0 ± 1.5 °C. After hatching, fry were maintained at the same conditions until the yolk sac resorption (approximately 2 weeks) [53]. After this stage, free-swimming fry (initial body weight of 0.31 ± 0.05 g) were randomly assigned to four experimental groups, according to the four experimental diets (CTRL, CTRL-ASX, A50, and A50-ASX) with 60 fish per group and 20 fish per tank (triplicate). Each experimental tank was equipped with an independent water inlet in a flow-through system that ensured a water renewal of 85%/h. Water parameters were as follows: temperature 14.0 ± 0.5 °C, dissolved oxygen 8.5 ± 0.5 mg/L, and pH 7.6. The daily amount of feed per tank was calculated as 5% of the fish’s total body weight, and it was provided by dividing it into 10 equal portions per day [54]. All the feed provided at each feeding time was completely consumed. The daily feed amount was adjusted every two weeks by weighing all the fish in each tank. Throughout the trial, fish were monitored daily, and any deceased specimens were promptly removed from the tanks and recorded for the calculation of survival rates. At the conclusion of the 60-day trial, following a 24 h fasting period, all fish were euthanized using a lethal dose of MS222 (0.3 g/L). The liver, intestine, and muscle tissues were then dissected and immediately stored for subsequent analyses.

### 2.7. Growth Indexes and Feed Conversion Ratio

At the end of the 60-day treatment period, all the fish from each tank (60 per experimental group) were weighed with an OHAUS Explorer analytical balance (Greifensee, Switzerland).

For each tank, relative growth rate (RGR), specific growth rate (SGR), and feed conversion ratio (FCR) were calculated as follows:RGR (%) = [(FBW − IBW)/IBW] × 100(1)SGR (% day^−1^) = [(ln FBW − ln IBW)/t] × 100(2)FCR = feed intake/weight gain(3)

FBW and IBW refer to final and initial body weights, respectively; t refers to days of trial.

Survival rate was calculated by removing the dead fish over the feeding trial from the initial number of fish per tank (at 60 days).

### 2.8. Chemical Digestion of Samples and MPs Quantification

Intestine, liver, and muscle samples were dissected from 5 rainbow trout fry per tank (15 fish for each experimental group) and frozen at −80 °C. According to the method described in Chemello et al. [55], each sample was weighed and placed in glass tubes. A 10% KOH solution with a ratio of 1:10 (weight/volume) was added to each of the tubes for the digestion. Tubes were incubated at 40 °C for 48 h. The digestate was then filtered through GF/F Whatman^®^ fiberglass filter papers (Merck KGaA, Darmstadt, Germany) with a pore size of 0.7 μm using a vacuum pump. The filters were dried at room temperature overnight and then stored in glass petri dishes. MPs quantification on filters was performed through a Zeiss Axio Imager.A2 (Zeiss, Oberkochen, Germany) using FITC (491 nm) and Texas Red (561 nm) channels. The MP were counted using the ZEN Blue 2.3 software (Zeiss) and the acquisition of images was made by the Axiocam 503 digital camera (Zeiss).

### 2.9. Histological Analysis

Intestine and liver samples of 5 rainbow trout fry per tank (15 fish for each experimental group) were fixed in Bouin’s solution (Merck KGaA) at 4 °C for 24 h and then stocked in a 70% ethanol solution. Samples were processed according to Zarantoniello et al. [56] to obtain sections of 5 μm in thickness that were stained with the following: (i) Mayer haematoxylin and eosin Y (Merck KGaA) to assess potential alterations in the tissues’ architecture and the eventual occurrence of inflammatory infiltration in both the intestinal tract and the hepatic parenchyma; (ii) Alcian Blue (Bio-Optica, Milan, Italy) to count the relative abundance of Alcian blue positive (Ab+) goblet cells in intestine sections. The ZEN 2.3 software (Zeiss) was used for the morphometric evaluation of the height of undamaged and non-oblique mucosal folds. Finally, for the assessment of the supranuclear vacuoles, a semi-quantitative scoring system was adopted, according to Pacorig et al. [57]. Scores were assigned as follows: 1, absent; 2, scattered; 3, diffused; 4, abundant; 5, highly abundant.

### 2.10. Real-Time qPCR

Intestine and liver samples of 3 rainbow trout per tank (9 fish per experimental group) were dissected and stored at −80 °C. Total RNA extraction, cDNA synthesis, and real-time qPCRs were conducted as described in Randazzo et al. [58]. The specificity of the qPCR products was determined by the melting curve that revealed one single peak for each run. Two no-template controls were loaded in each PCR run, and they did not show peaks in each run. Using beta-actin (*β-actin*) and 60S ribosomal protein (*60s*) as housekeeping genes, relative quantification of interleukin-1 (*il1*), interleukin-10 (*il10*), and tumor necrosis factor alpha (*tnfa*) was performed on intestine samples, whilst that of superoxide dismutase 1 (*sod1*), superoxide dismutase 2 (*sod2*), and catalase (*cat*) was performed on liver samples. Primer sequences are shown in Table 1.

### 2.11. Statistical Analysis

For growth index data, tanks were used as experimental units, whilst fish were the experimental units for all the remaining analyses. For the normality and homoscedasticity of all the data, Shapiro–Wilk and Levene’s tests were used, respectively. The data were analyzed using an Analysis of Variance (ANOVA), followed by Tukey’s post-hoc test, performed using the software package Prism 8 (GraphPad software version 8.0.2, San Diego, CA, USA). The obtained results were evaluated at a *p* < 0.05 significance level.

## 3. Results

### 3.1. Growth Rates

Based on calculations using the initial and final body weights of the fish, no significant differences were observed among the groups in terms of SGR, RGR, and FCR. The indexes’ values for the experimental groups are provided in Table 2.

### 3.2. MP Quantification

MP amounts determined in the intestine, liver, and muscle of rainbow trout are presented in Table 3. Fluorescent MP microbeads were not detected in both CTRL and CTRL-ASX. Considering fish fed MP contaminated diets, the A50-ASX group was characterized by a significantly (*p* < 0.05) lower MP abundance in both intestine and liver samples compared to the A50 one. Differently, no significant differences were detected between the A50 and A50-ASX groups in muscle samples.

For both A50 and A50-ASX, the number of MP in muscle tissue was found to be the lowest compared to liver and intestine. In addition, although it was not statistically significant, the MP amount detected in the A50-ASX group was lower compared to A50.

### 3.3. Histology

No pathological alterations or signs of inflammation were observed in either the intestine or the liver in any experimental group. The proper architecture of the organs was not altered by different dietary treatments. Considering the histological indexes evaluated in the intestine (Table 4), no significant differences were found among the experimental groups in terms of mucosal fold height. Differently, A50 and A50-ASX groups were characterized by a significant (*p* < 0.05) increase in the enterocytes’ supranuclear vacuolization (Figure 1d,e) and the relative abundance of Ab+ goblet cells (Figure 1c,f) compared to both CTRL and CTRL-ASX ones which did not show significant differences between them for both these indexes.

Considering the liver, all experimental groups exhibited a physiological liver parenchyma without signs of inflammation. Fish from all groups showed moderate fatty liver parenchyma, with widespread hepatocytes containing fat-filled cytoplasm (Figure 2). Regarding the fat fraction percentage, the A50 group exhibited a significantly (*p* < 0.05) higher value compared to the other experimental groups. The mean fat fraction (%) in liver tissues was as follows: 47.81 ± 5.97, 44.29 ± 4.54, 55.66 ± 8.35, and 43.26 ± 5.06 for CTRL, CTRL-ASX, A50, and A50-ASX, respectively.

### 3.4. Real-Time PCR

Regarding the oxidative stress response (Figure 3a–c), the A50 group exhibited significantly (*p* < 0.05) higher gene expression levels of *sod1*, *sod2*, and *cat* compared to the CTRL and CTRL-ASX groups. Differently, fish from the A50-ASX group were characterized by a *sod1*, *sod2*, and *cat* gene expression comparable to that detected in both control groups and significantly (*p* < 0.05) lower with respect to the A50 group. Considering the expression of immune-related markers (*il1b*, *il10*, and *tnfα*; Figure 3d–f), no significant differences were found among the experimental groups.

## 4. Discussion

The MP presence in aquafeeds represents a common route of contamination in farmed fish species [59,60]. Research indicates that the adverse effects of MP, as well as their translocation, are influenced by factors such as size, treatment duration, concentration, and polymer composition of the MP, along with the fish species, life stage, and feeding behavior [15,61,62,63].

The development of natural strategies to mitigate the negative effects of MP exposure on fish welfare is crucial for the sustainability of aquaculture. This study aimed to evaluate the efficacy of dietary microencapsulated ASX in mitigating MP accumulation and associated adverse effects in rainbow trout fry. Fish from all experimental groups readily consumed the respective feed at each feeding time, ensuring a 100% feed intake. In addition, the provision of all the experimental diets, prepared starting from the same batches of ingredients and following a commercial standard formulation for rainbow trout fry, resulted in the absence of significant differences in FCR at the end of the trial. Accordingly, no significant differences were detected among experimental groups in terms of both SGR and RGR, confirming results obtained in previous studies which did not highlight adverse effects of dietary MPs on growth performances of European seabass [23,64], yellow perch (*Perca flavescens*; [11]), three-spined stickleback (*Gasterosteus aculeatus*; [65]), and gilthead seabream (*Sparus aurata*; [66]). Furthermore, none of the groups fed diets containing MPs exhibited signs of intestinal inflammation. This was corroborated by both histological indexes and the relative expression of immune response markers (*il1b*, *il10*, and *tnfα*), suggesting that the transit of small MPs (1–5 µm, as used in this study) did not affect intestinal structure or functionality, as previously demonstrated in various fish species [16,23,67]. In contrast, more severe intestinal alterations have been observed in fish exposed to MPs with a larger size range [14,64,68]. Notably, both the A50 and A50-ASX groups exhibited an increase in goblet cell abundance relative to the control, which may be explained by MP impairing the mucus barrier properties [67], prompting an increase in goblet cells to counteract this effect. Additionally, dietary supplementation of 50 mg/kg of MPs with a size range of 1–5 µm resulted in effects similar to those observed in zebrafish and European seabass exposed to the same fluorescent microbeads [15,16,23].

In this study, MPs were found to have the highest concentration in the liver, followed by the intestine and muscle in both the A50 and A50-ASX groups. This pattern corresponds with previous findings that dietary MPs of the same size range (1–5 µm) can be absorbed at the intestinal level and subsequently translocated to other organs, primarily the liver, through the bloodstream [16,17,18,19].

Considering absorption, it has been demonstrated that enterocytes’ supranuclear vacuoles are involved in the internalization of dietary MPs, following the same absorption route as intact nutrient molecules via pinocytosis [69,70]. Accordingly, in the present study, the density of supranuclear vacuoles was significantly higher in both the A50 and A50-ASX groups compared to the control groups. As suggested by the MP quantification analysis, once absorbed, MPs were translocated to the liver, leading to substantial accumulation within the hepatic tissue and preventing their further translocation to other tissues, including muscle. This accumulation led to detrimental effects on liver health. In fact, fish from the A50 group exhibited significantly higher hepatic lipid accumulation compared to other experimental groups. These findings are consistent with previous research showing that MPs translocated to the liver can disrupt lipid signaling pathways, resulting in lipid accumulation and potentially leading to steatosis [22]. Additionally, as demonstrated in previous studies, one of the most detrimental effects of MP exposure in fish is oxidative stress, which may adversely affect the overall health and quality of farmed fish over time [68,71,72,73]. Accordingly, in the present study, a significantly higher expression of genes related to the oxidative stress response (*sod1*, *sod2*, and *cat*) was evidenced in the liver samples from the A50 group. Interestingly, fish from the A50-ASX group exhibited expression levels of *sod1*, *sod2*, and *cat* that were comparable to those observed in both control groups. This outcome can be attributed to the provision of a potent antioxidant, such as natural ASX, which has been shown to effectively mitigate MP-induced oxidative stress [16,23]. Additionally, the reduction in oxidative stress in this group may also be linked to a lower MP abundance in the liver, which likely resulted in milder adverse effects, as evidenced by the absence of significant differences in hepatic lipid accumulation compared to the control groups.

Finally, the microencapsulation technology employed to deliver and preserve natural ASX played a crucial role in reducing MP absorption at the intestinal level, thereby decreasing the number of MPs translocated to other organs, including the liver. Specifically, the wall matrix of the microcapsules was primarily composed of starch, which is known to possess coagulation properties for MPs, particularly enhanced in the acidic conditions of the fish stomach [23,74]. Microcapsules, after releasing their contents, were thus able to coagulate MP microbeads in the gut, forming coagula that were too large to be absorbed by enterocytes. This result was further confirmed by the lower density of enterocytes’ supranuclear vacuoles in the A50-ASX group compared to the A50 group. Similar findings were reported in studies using this technology in response to dietary contamination with fluorescent MP microbeads of 1–5 µm in size [16,23]. In European seabass juveniles, microencapsulation technology significantly reduced the absorption of MPs at a 50 mg/kg dietary concentration over a 2-month feeding trial, similar to the present study [23]. In contrast, in zebrafish, the same technology was less effective in preventing the MP absorption at a 50 mg/kg dietary concentration but showed promising results when a higher MP concentration (500 mg/kg) was used [16]. This discrepancy can be attributed to the differences in the gut environments of the fish species involved.

Starch has been reported to initiate the coagulation phenomenon by bridging and entangling suspended particles under acidic conditions [74,75], such as those found in the stomachs of carnivorous fish. In contrast, zebrafish, like other cyprinids, lack a stomach and do not possess a strong acidic digestive environment [76]. Therefore, it can be hypothesized that in zebrafish fed the 50 mg/kg MP and ASX diet, the low MP concentration in the digestive tract, combined with the absence of a highly acidic environment, was insufficient to trigger coagulation processes. Consequently, the success of the present study can also be attributed to the fact that rainbow trout, like European seabass, are carnivorous species with an acidic stomach, allowing the coagulating agents in the ASX encapsulation technology to function more effectively.

Rainbow trout, the primary freshwater species cultivated in the EU [77], are commonly fed ASX-supplemented diets during the finishing period to achieve the characteristic red-to-pink flesh coloration typical of salmonids [78]. Given that most farmed fish in aquaculture are carnivores, the microencapsulation technology used to deliver ASX, along with its influence on pigmentation, presents a valuable mechanism for promoting fish well-being.

## 5. Conclusions

Dietary MPs caused oxidative stress and hepatic alterations in the A50 group, with the liver being the main accumulation site of feed-derived MPs. Microencapsulated ASX effectively mitigated these adverse effects in the A50-ASX group since: (i) ASX’s natural antioxidant properties helped to reduce oxidative stress at the hepatic level; (ii) the materials composing the microcapsule walls decreased MP absorption at the intestinal level, facilitating the formation of coagula too large for absorption. This technique is especially effective in carnivorous fish like rainbow trout due to stomach acidity, which was shown to enhance dietary MP coagulation. This is a promising approach for the aquaculture industry, which, however, requires further investigation across various species and conditions to fully explore its potential against dietary MP contamination.

## Figures and Tables

**Figure 1 animals-15-01020-f001:**
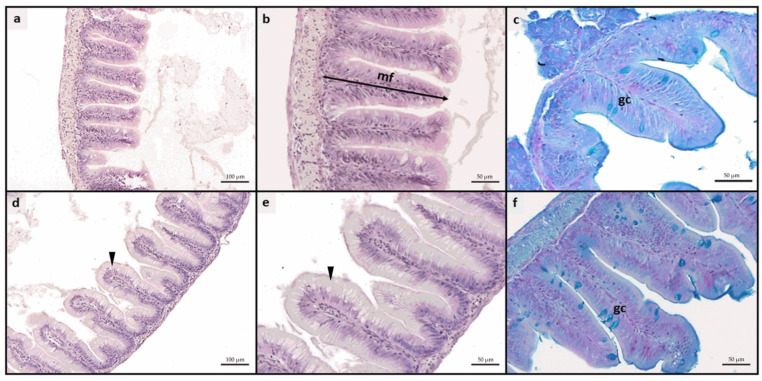
Example of histomorphology of intestine rainbow trout fry. (**a**–**c**) mucosal architecture of fish fed CTRL diet; (**d**–**f**) details of mucosal architecture of fish fed A50 diet, characterized by a significant increase in enterocytes’ supranuclear vacuolization (arrowhead) and relative abundance of Ab+ goblet cells. CTRL, fish fed control diet; CTRL-ASX, fish fed control diet implemented with 7 g/kg of microencapsulated ASX; A50, fish fed control diet including 50 mg/kg of fluorescent MP; A50-ASX, fish fed control diet including 50 mg/kg of fluorescent MP and implemented with 7 g/kg of microencapsulated ASX. Abbreviations: mf, mucosal folds; gc, goblet cells. Scale bars: (**a**,**d**) 100 µm; (**b**,**c**,**e**,**f**) 50 µm.

**Figure 2 animals-15-01020-f002:**
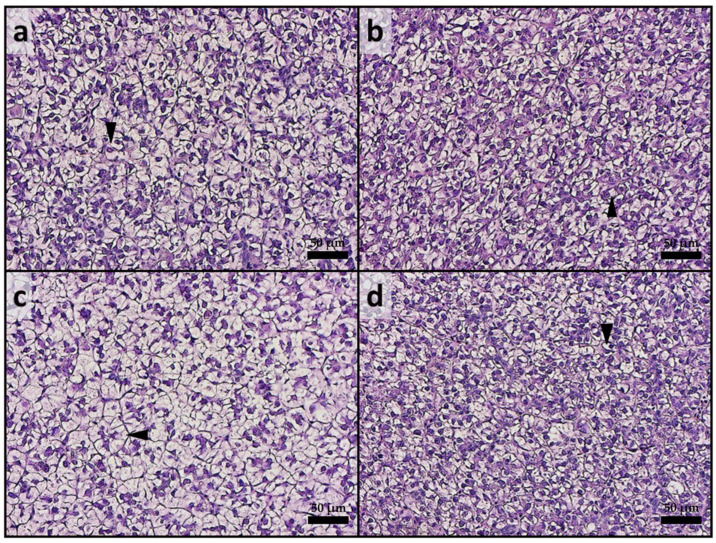
Example of histomorphology of liver from rainbow trout fry from the different experimental groups. (**a**) CTRL; (**b**) CTRL-ASX; (**c**) A50, characterized by a higher incidence of fat-filled hepatocytes (arrowhead); (**d**) A50-ASX. CTRL, fish fed control diet; CTRL-ASX, fish fed control diet implemented with 7 g/kg of microencapsulated ASX; A50, fish fed control diet including 50 mg/kg of fluorescent MP; A50-ASX, fish fed control diet including 50 mg/kg of fluorescent MP and implemented with 7 g/kg of microencapsulated ASX. Scale bars: 50 µm.

**Figure 3 animals-15-01020-f003:**
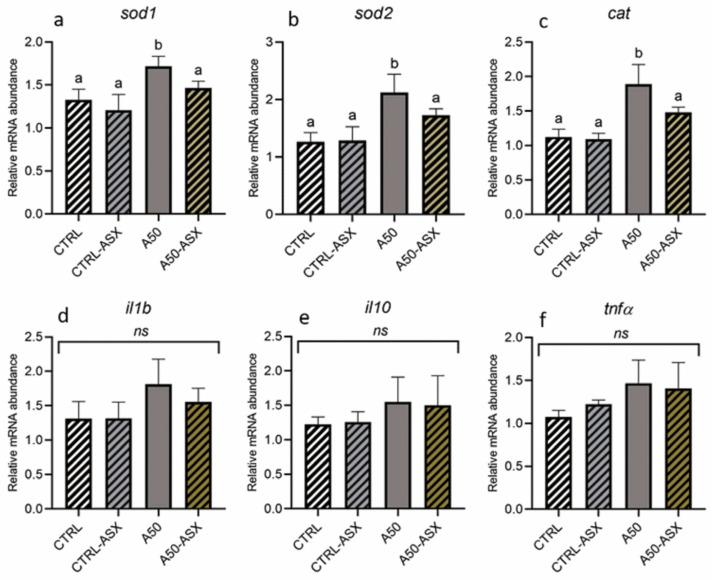
Relative mRNA abundance of genes involved in (**a**–**c**) the oxidative stress response analyzed in the liver and (**d**–**f**) the immune response analyzed in the intestine of rainbow trout fry from the different experimental groups. Data are presented as mean ± standard deviation (*n* = 9). ^a,b^ different letters denote statistically significant differences among the experimental groups. CTRL, fish fed control diet; CTRL-ASX, fish fed control diet implemented with 7 g/kg of microencapsulated ASX; A50, fish fed control diet including 50 mg/kg of fluorescent MP; A50-ASX, fish fed control diet including 50 mg/kg of fluorescent MP and implemented with 7 g/kg of microencapsulated ASX. Abbreviations: ns, no significant differences (*p* > 0.05).

**Table 1 animals-15-01020-t001:** Sequences and identification numbers of primers used in the current study.

Genes	Forward Sequence (5′-3′)	Reverse Sequence (5′-3′)	NCBI ID
*il1b*	GCTGGGGATGTGGACTTC	GTGGATTGGGGTTTGATGTG	040702-2
*il10*	ATTTGTGGAGGGCTTTCCTT	AGAGCTGTTGGCAGAATGGT	051111-1
*tnfa*	TTGTGGTGGGGTTTGATG	TTGGGGCATTTTATTTTGTAAG	050317-1
*sod1*	AACCATGGTGATCCACGAGA	ATGCCGATGACTCCACAGG	FJ_860004.1
*sod2*	TGCCCTCCAGCCTGCTCT	CTTCTGGAAGGAGCCAAAGTC	MH_138007.1
*cat*	GGCTGGGAGCCAACTATCTG	GGAGCTCCACCTTGGTTGTC	MH_138006.1
*60S* (hk)	GGTACCCATCTCCTGCTCCAA	GACGTCGCACTTCATGATGCT	AJ_537421
*b-actin* (hk)	AGACCACCTTCAACTCCATCAT	AGAGGTGATCTCCTTCTGCAT	AJ537421

Abbreviations: hk, housekeeping genes.

**Table 2 animals-15-01020-t002:** Growth indexes and feed conversion ratio calculated for rainbow trout fry from the different experimental groups.

	CTRL	CTRL-ASX	A50	A50-ASX	*p*-Value
SGR (% day^−1^)	3.2 ± 0.6	3.4 ± 0.5	3.3 ± 0.6	3.5 ± 0.4	0.9090
RGR (%)	627.8 ± 217.1	687.7 ± 219.1	687.0 ± 245.0	759.4 ± 192.4	0.9063
FCR	1.10 ± 0.03	1.07 ± 0.03	1.05 ± 0.04	1.05 ± 0.03	0.2734

Values are shown as mean ± SD (*n* = 3). CTRL, fish fed control diet; CTRL-ASX, fish fed control diet implemented with 7 g/kg of microencapsulated ASX; A50, fish fed control diet including 50 mg/kg of fluorescent MP; A50-ASX, fish fed control diet including 50 mg/kg of fluorescent MP and implemented with 7 g/kg of microencapsulated ASX. Abbreviations: SGR, specific growth rate; RGR, relative growth rate; FCR, feed conversion ratio.

**Table 3 animals-15-01020-t003:** Quantification of amino formaldehyde polymer (FMv-1.3) fluorescent MP microbeads in samples of intestine, liver, and muscle of rainbow trout fry from the different experimental groups (microbeads/mg of fresh tissue).

	CTRL	CTRL-ASX	A50	A50-ASX	*p*-Value
Intestine	0	0	9.1 ± 2.1 ^a^	3.3 ± 0.9 ^b^	<0.0001
Liver	0	0	40.0 ± 4.5 ^a^	10.9 ± 1.4 ^b^	<0.0001
Muscle	0	0	1.0 ± 0.9 ^a^	0.5 ± 0.3 ^a^	<0.0001

Values are shown as mean ± standard deviation (*n* = 15 for each tissue). ^a,b^ Within each line, different letters denote statistically significant differences among the experimental group. CTRL, fish fed control diet; CTRL-ASX, fish fed control diet implemented with 7 g/kg of microencapsulated ASX; A50, fish fed control diet including 50 mg/kg of fluorescent MP; A50-ASX, fish fed control diet including 50 mg/kg of fluorescent MP and implemented with 7 g/kg of microencapsulated ASX.

**Table 4 animals-15-01020-t004:** Histological indexes measured in the intestine of rainbow trout fry from the different experimental groups.

	CTRL	CTRL-ASX	A50	A50-ASX	*p*-Value
Mucosal fold height	225.5 ± 40.4	225.8 ± 41.2	241.3 ± 41.6	226.9 ± 32.7	0.6362
Supranuclear vacuoles (score)	1.67 ± 0.49 ^a^	1.93 ± 0.88 ^a^	4.67 ± 0.49 ^c^	3.27 ± 0.46 ^b^	<0.0001
Ab+ goblet cells per fold	4.77 ± 0.83 ^a^	5.08 ± 0.76 ^a^	8.46 ± 1.13 ^b^	8.54 ± 1.33 ^b^	<0.0001

Data are reported as mean ± standard deviation (*n* = 15). ^a–c^ Within each line, different letters denote statistically significant differences among the experimental group. Scores for supranuclear vacuoles were assigned as follows: 1, absent; 2, scattered; 3, diffused; 4, abundant; 5, highly abundant. CTRL, fish fed control diet; CTRL-ASX, fish fed control diet implemented with 7 g/kg of microencapsulated ASX; A50, fish fed control diet including 50 mg/kg of fluorescent MP; A50-ASX, fish fed control diet including 50 mg/kg of fluorescent MP and implemented with 7 g/kg of microencapsulated ASX.

## Data Availability

The data presented in the current study are available from the corresponding author on a reasonable request.

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
