# Peer review of "Mitigation of Dietary Microplastic Accumulation and Oxidative Stress Response in Rainbow Trout (Oncorhynchus mykiss) Fry Through Dietary Supplementation of a Natural Microencapsulated Antioxidant"

_animals, 2025, doi:10.3390/ani15071020_

Round 1
Reviewer 1 Report
Comments and Suggestions for Authors
The study presents a timely investigation into the potential of microencapsulated astaxanthin (AX) to mitigate the harmful effects of dietary microplastics (MPs) in aquaculture species, specifically rainbow trout fry (Oncorhynchus mykiss). The authors effectively highlight the environmental and health concerns associated with MP contamination and propose a biotechnological approach to reduce oxidative stress and MP absorption in fish. However, several notable limitations need to be addressed:
- Experimental Dose Justification: The rationale behind selecting 50 mg MPs/kg as the experimental dose is not adequately justified. It remains unclear whether this concentration reflects levels commonly found in commercial aquafeeds or if it was chosen arbitrarily.
- Microplastic Characterization: More information is needed regarding the nature of the MPs used in the study, including their purity, presence of additives, and polymer types. This is crucial for understanding their potential interactions with AX and biological systems.
- Sample Size and Statistical Analysis: The use of only n = 3 for growth indexes is insufficient for robust analysis.
- Drying Process Clarification: The statement "the filters were dried at room temperature and stored in glass petri dishes" should specify the duration of the drying process.
- Growth and Feed Efficiency: While the study reports unaffected growth despite oxidative stress, a more detailed explanation is needed, including additional growth indicators such as feed conversion ratio and feed intake. This would help to better interpret the physiological responses observed.
- Feed Intake and MP Intake: The reduction of microplastics in samples from the intestine, liver, and muscle of rainbow trout (A50-ASX) is not clearly because it may be Juste related to a reduction in feed intake consequence in MP intake.
- MP Quantification in Table 3: The MP quantification in samples of the intestine, liver, and muscle of rainbow trout fry from the different experimental groups (microbeads/mg) should specify whether the measurements are based on fresh or dry weight, or another standard.
- Microplastic Quantification: The absence of microplastic quantification in the control groups (CTRL and CTRL-ASX) raises concerns regarding the potential presence of microplastics in the experimental environment, which could influence the study's findings.
- Effect of MPs and AX on Organ Weight: Does the presence of MPs and AX affect the weight of the intestine, liver, and muscle of rainbow trout? Please clarify the absolute number of MPs in these organs. Is the decrease in MP/ mg organs related can be result of increased tissue mass?
- Organ-specific Analysis: The study assesses MP accumulation in the intestine, liver, and muscle but does not provide a clear rationale for the selection of these organs. It is unclear why other key organs, such as the kidneys, brain, and gills, were not analyzed.
- Statistical Analysis: Including p-values in all tables and replacing qualitative scoring (e.g., +++) with exact numerical values and standard deviations in histological data (Table 4).
- Figure Clarity: Figures 1 and 2 should be presented with clearer differentiation and detailed legends to illustrate differences more precisely, ensuring their clarity and relevance to the study's findings.
Author Response
The study presents a timely investigation into the potential of microencapsulated astaxanthin (AX) to mitigate the harmful effects of dietary microplastics (MPs) in aquaculture species, specifically rainbow trout fry (Oncorhynchus mykiss). The authors effectively highlight the environmental and health concerns associated with MP contamination and propose a biotechnological approach to reduce oxidative stress and MP absorption in fish. However, several notable limitations need to be addressed:
1. Experimental Dose Justification: The rationale behind selecting 50 mg MPs/kg as the experimental dose is not adequately justified. It remains unclear whether this concentration reflects levels commonly found in commercial aquafeeds or if it was chosen arbitrarily.
Authors’ reply: The reviewer is right and possibly the rationale behind the choice of this MP concentration was not clearly explained. We previously used this concentration as representative of the intrinsic contamination of fish feeds accordingly to the available literature. In addition, we previously demonstrated that this concentration (used in comparison to the 10x higher one) was not sufficient to trigger coagulation processes in the zebrafish gut (a freshwater fish that lack a true acid digestion). Differently in European seabass (saltwater fish with stomach) MP were able to coagulate when provided at 50mg/kg due to the effect of the stomach’ acidic environment on the coagulative properties of starch present in the AX microcapsules’ wall matrix.
For these reasons, the aim of the present study was to verify if this low MP concentration was sufficient to trigger coagulation processes even in rainbow trout fry, a freshwater fish with stomach in an early stage of its life cycle. We better clarified this concept in the introduction section.
2. Microplastic Characterization: More information is needed regarding the nature of the MPs used in the study, including their purity, presence of additives, and polymer types. This is crucial for understanding their potential interactions with AX and biological systems.
Authors’ reply: MP chosen for the present study were the same selected for the previous studies on the same topic, and thus: (i) able to be absorbed at intestinal level and translocate to other tissues; (ii) able to induce an oxidative stress response when accumulated in the liver; (iii) prone to be coagulated by the AX microcapsule. To compare the results obtained in the present study with the previous ones, we wanted to replicate their experimental conditions using commercially available fluorescent microbeads certified for their characteristics and purity. In light to provide more details as possible regarding the MP used, the link to the data shit has been added in the 2.2 section of Materials and Methods.
3. Sample Size and Statistical Analysis: The use of only n = 3 for growth indexes is insufficient for robust analysis
Authors’ reply: In the present study, all the fish were weighted to determine the final weight of each specimen. However, for growth indexes, the use of a n that is equal to the number of tanks is a common practice in aquaculture studies. In fact, fish growth is subjected to environmental dependence (water quality parameters, water circulation, etc) and individual interaction effects (social hierarchies, competition). These interactions create a collective effect on growth that is better depicted by considering the tank as a whole. This approach provides a more accurate assessment of treatment effects on the farmed fish. (https://doi.org/10.1016/S0044-8486(00)00542-1; https://doi.org/10.1016/j.aquaculture.2015.05.018; https://doi.org/10.3390/ani12131698; https://doi.org/10.1155/2023/3465335)
4. Drying Process Clarification: The statement "the filters were dried at room temperature and stored in glass petri dishes" should specify the duration of the drying process.
Authors’ reply: The reviewer is right, and the duration of the drying process has been added to the section.
5. Growth and Feed Efficiency: While the study reports unaffected growth despite oxidative stress, a more detailed explanation is needed, including additional growth indicators such as feed conversion ratio and feed intake. This would help to better interpret the physiological responses observed.
Authors’ reply: As suggested also by the Reviewer 2, feed conversion ratio and feed intake have been added to the MS. Particularly, all the feed provided at each feeding was completely consumed by the fish, ensuring a 100% of feed intake. This information has been added to the experimental design section. Differently, FCR methods and results have been added to the dedicated sections.
6. Feed Intake and MP Intake: The reduction of microplastics in samples from the intestine, liver, and muscle of rainbow trout (A50-ASX) is not clearly because it may be Juste related to a reduction in feed intake consequence in MP intake.
Authors’ reply: Feed intake was the same for each tank. In fact, rainbow trout fry are active feeders that readily consume the pellet, especially because the daily ration was divided in 10 portions per day which ensured a proper access to feed to all the specimens. This result was supported by the absence of significant differences in growth indexes. This can be explained by the fact that all the experimental diets were produced following the same formulation (thus did not differ in terms of nutritional composition) and thus were able to satisfy all the fish nutritional requirements.
7. MP Quantification in Table 3: The MP quantification in samples of the intestine, liver, and muscle of rainbow trout fry from the different experimental groups (microbeads/mg) should specify whether the measurements are based on fresh or dry weight, or another standard.
Authors’ reply: The reviewer is right; measurements were based on fresh weight of the organs. This detail has been added to the Table 3 caption.
8. Microplastic Quantification: The absence of microplastic quantification in the control groups (CTRL and CTRL-ASX) raises concerns regarding the potential presence of microplastics in the experimental environment, which could influence the study's findings.
Authors’ reply: The aim of the present study was to exclusively assess the effects of voluntarily added fluorescent MP microbeads during the preparation of the experimental diets. Thus, we excluded the potential intrinsic contamination of the ingredients used or the MP present in the environment during the trial. Since the basic mixture for diets’ production was composed of the same ingredients from the same batches and the environmental conditions to which fish were exposed were the same for each tank, it can be assumed that the level of potential “external” MP contamination was consistent among all the experimental group. Therefore, it can be stated that any effect observed in fish is uniquely attributable to the added fluorescent MP microbeads.
9. Effect of MPs and AX on Organ Weight: Does the presence of MPs and AX affect the weight of the intestine, liver, and muscle of rainbow trout? Please clarify the absolute number of MPs in these organs. Is the decrease in MP/ mg organs related can be result of increased tissue mass?
Authors’ reply: The presence of MP or AX did not affect the weight of the organs that present only a variability due to physiological differences among specimens. The whole organs were weighted and digested so we obtained the absolute number of MP. However, these types of results are expressed as number of MP / mg of tissue in light to avoid differences in tissues’ mass due to individual variability and to allow comparison among specimens, experimental groups and also different studies.
10. Organ-specific Analysis: The study assesses MP accumulation in the intestine, liver, and muscle but does not provide a clear rationale for the selection of these organs. It is unclear why other key organs, such as the kidneys, brain, and gills, were not analyzed.
Authors’ reply: The rationale was described in the introduction section. These organs were selected because directly affected by the MPs contained in the diet. In fact, intestine is directly involved in the absorption of MP that are introduced with the diets and liver is the major organ in which absorbed dietary-derived MP are accumulated (causing oxidative stress). Finally, assessing their translocation (or a reduction in the translocation) in the muscle is important for the product quality and safety in aquaculture. In addition, the crucial point of the present study was not the effect of dietary MP (which was already demonstrated by a wide scientific literature) but rather the effects of the microencapsulated AX on the MP. Other organs are more affected to the MP presence in the surrounding water (like gills) or are interested only by accumulation of smaller MP (nanoplastics in the case of brain) compared to those used in the present study. For that reason, other organs were not the target of the present study, so they were not considered.
11. Statistical Analysis: Including p-values in all tables and replacing qualitative scoring (e.g., +++) with exact numerical values and standard deviations in histological data (Table 4)
Authors’ reply: p-values have been added to all tables. Regarding histological analyses, qualitative scoring is used because some indexes (including the degree of supranuclear vacuolization of enterocytes and the presence of inflammatory events) are not directly measured and are the results of the frequency of which these events occur. Following the Reviewer’s suggestion, results of histological indexes have been reported as follows: (i) since no inflammatory events were detected in fish from all the experimental groups (as evidenced by the – score in all the groups), we have substituted the index in Table 4 with a general sentence in the Results section (referred to both intestine and liver due to the absence of pathological lymphocytes infiltration in these organs); (ii) for supranuclear vacuoles, we converted the qualitative scoring (represented by +) to numerical values on which a statistical analysis has been conducted; the semi-quantitative scoring system associate a numeric value (from 1 to 5) to histological observations (from absent to highly abundant). This scoring system is a very widespread method for the assessment of histopathological changes in fish (https://doi.org/10.2174/97816810858071170101); (iii) we reported the relative abundance of Ab+ goblet cells as number of cells counted per fold instead of an index.
Methods and Results sections regarding histology have been modified accordingly.
12. Figure Clarity: Figures 1 and 2 should be presented with clearer differentiation and detailed legends to illustrate differences more precisely, ensuring their clarity and relevance to the study's findings.
Authors’ reply: Thank you for the suggestion; figures’ captions have been accordingly revised.
Reviewer 2 Report
Comments and Suggestions for Authors
The MS "Mitigation of dietary microplastics accumulation and oxidative
stress response in rainbow trout (Oncorhynchus mykiss) fry
through dietary supplementation of a natural microencapsulated antioxidant" evaluated the effectiveness of a microencapsulated natural antioxidant, astaxanthin (AX), in mitigating the adverse effects of dietary MPs in rainbow trout fry. This is an excellent piece of work to address the menaces of microplastics in Aquaculture system. Here are some suggestion needs to be carried out:
Line 102-110 complicates the objective of the MS and authors are suggested to rephrase and rewrite these lines to make it simple so that hypothesis and objectives of the study clear and contrast.
Authors are suggested to include a paragraph about rainbow trout (Oncorhynchus mykiss) and why they have chosen this fish for this research.
Why did you choose "Amino formaldehyde polymer (FMv-1.3) fluorescent MP"?
Authors stated that " The detailed process of microcapsule preparation is protected under intellectual property rights" but replication of such studies is not possible without complete methodology".Authors are instructed to furnish the complete methodology for microcapsule preparation.
Authors states that "Due to the chemical properties of the microcapsules, their contents were released approximately 90 seconds after immersion in water". However, it is not always possible that such feed will be accepted by the fishes within such less duration. Authors should provide the data on nutrient leaching especially of astaxanthin (AX).
Authors need to provide the husbandry condition of fish along with experimental trail details.
Authors need to furnish full details "The daily feeding rate was set at 5% of the fish total body weight, divided in 10 equal portions per day" .
Authors need to provide the feed intake data along with FCR.
Authors are also suggested to give full details of the formulated feed with ingredients and additives.
Results and discussion section is nicely written with lot of supporting data.
Conclusion section needs to be rewritten in short and precise manner.
Author Response
The MS "Mitigation of dietary microplastics accumulation and oxidative
stress response in rainbow trout (Oncorhynchus mykiss) fry
through dietary supplementation of a natural microencapsulated antioxidant" evaluated the effectiveness of a microencapsulated natural antioxidant, astaxanthin (AX), in mitigating the adverse effects of dietary MPs in rainbow trout fry. This is an excellent piece of work to address the menaces of microplastics in Aquaculture system. Here are some suggestion needs to be carried out:
Line 102-110 complicates the objective of the MS and authors are suggested to rephrase and rewrite these lines to make it simple so that hypothesis and objectives of the study clear and contrast.
Authors’ reply: In these lines, a new sentence has been added to clarify the rationale behind the use of 50 mg/kg for MP (as suggested by Reviewer 1) and some parts have been rewritten according to the suggestion.
Authors are suggested to include a paragraph about rainbow trout (Oncorhynchus mykiss) and why they have chosen this fish for this research.
Authors’ reply: A sentence has been added in the introduction to justify the choice.
Why did you choose "Amino formaldehyde polymer (FMv-1.3) fluorescent MP"?
Authors’ reply: Our choice was oriented to commercially available MP microbeads certified for their features and with particular excitation and emission wavelengths necessary to clearly discriminate them respect to biological tissues analysed (fish samples). At the time of purchasing, only the amino formaldehyde microbeads were available on the company website with these features and, overall, with the range size 1-5 µm. The suitability of this MP for our purpose was certified by the use of the same fluorescent microbeads in previous studies on the same topic (Cattaneo et al. 2023, 2024; Zarantoniello et al. 2024). In addition, the link to the data shit has been added in the 2.2 section of Materials and Methods, also according to a comment from Reviewer 1.
Authors stated that " The detailed process of microcapsule preparation is protected under intellectual property rights" but replication of such studies is not possible without complete methodology". Authors are instructed to furnish the complete methodology for microcapsule preparation.
Authors’ reply: We provide as much details as possible to illustrate the features of the microcapsules used in the present study, which can be considered sufficient to understand the applicability of this technology. Unfortunately, as product from a company, we are not able to fully disclose the methodology for their preparation. This aspect is also included in the text and was also accepted in previously published manuscripts.
Authors states that "Due to the chemical properties of the microcapsules, their contents were released approximately 90 seconds after immersion in water". However, it is not always possible that such feed will be accepted by the fishes within such less duration. Authors should provide the data on nutrient leaching especially of astaxanthin (AX).
Authors’ reply: The reviewer is right since the rate of feed consumption by the fish depends on several factors, including environmental features, feed features (pellet size and type), and fish behaviour, health status, and stocking density. For the present study, fish maintenance was daily monitored to avoid alterations in water parameters, and we adopted the official guidelines regarding the pellet size and feeding frequency for rainbow trout fry. We are not able to provide data on nutrient leaching regarding our experiment. However, it should be considered that rainbow trout fry are active feeders that readily consume appropriately sized pellet. In addition, the high frequency of feeding suggested for fry and adopted for the present trial (10 portions per day) ensured that all the specimens had access to feed and the individual consumption of a relatively small number of pellets is quite rapid, likely within few seconds once a fry encounters a pellet. Finally, this microencapsulation technology has been deeply tested by the company to guarantee an optimal release of the microcapsule’s content (ASX in this case).
Authors need to provide the husbandry condition of fish along with experimental trail details.
Authors’ reply: More details regarding the fish rearing conditions have been added.
Authors need to furnish full details "The daily feeding rate was set at 5% of the fish total body weight, divided in 10 equal portions per day" .
Authors’ reply: The daily amount of feed per tank was calculated basing on the 5% of the fish total body weight and it was provided dividing it in 10 equal portions per day. The daily amount of feed was adjusted every two weeks by weighing all the fish in each tank. These details have been added in the Experimental design section.
Authors need to provide the feed intake data along with FCR.
Authors’ reply: All the feed provided at each feeding was completely consumed by the fish, ensuring a 100% of feed intake. This information has been added to the experimental design section. Differently, FCR methods and results have been added to the dedicated sections.
Authors are also suggested to give full details of the formulated feed with ingredients and additives.
Authors’ reply: The dietary formulation was not added since fish from all the experimental groups were fed diets with the same dietary formulation. In fact, all the diets were prepared starting from the same batches of ingredients to resemble a commercial standard diet for rainbow trout fry used in a previous study (https://doi.org/10.1016/j.aquaculture.2021.736446). The reference to this work was mistakenly not reported in the sentence regarding experimental diets.
Results and discussion section is nicely written with lot of supporting data.
Authors’ reply: thank you for your comment.
Conclusion section needs to be rewritten in short and precise manner.
Authors’ reply: Conclusion section has been rewritten according to the suggestion.
Round 2
Reviewer 1 Report
Comments and Suggestions for Authors
- Below the table and figure, indicate all the abbreviations used, for example: CTRL, CTRL-ASX, A50, A50-ASX
- Table 3 specify MP polymers used in title.
- Abstract and Title: specify MP polymers used
- Discussion Section: Add information regarding the feed conversion ratio.
- Use of abbreviation: If abbreviations are defined once, do not repeat their definitions. For example, in line 213, specific growth rate (SGR), relative growth rate (RGR), and feed conversion ratio (FCR) are defined. From line 269 onwards, use SGR, RGR, and FCR without redefining them.
Author Response
- Below the table and figure, indicate all the abbreviations used, for example: CTRL, CTRL-ASX, A50, A50-ASX
Authors’ reply: Abbreviations have been added
- Table 3 specify MP polymers used in title.
Authors’ reply: The polymer has been specified in Table 3
- Abstract and Title: specify MP polymers used
Authors’ reply: The type of polymer has been specified in the abstract. To avoid weighing down the article title, we preferred not to specify the type of polymer here and keep it more general. As also explained to the Reviewer 2 during Round 1, our choice was oriented to commercially available MP microbeads certified for their features and with particular excitation and emission wavelengths necessary to clearly discriminate them respect to biological tissues analysed (fish samples). At the time of purchasing, only the amino formaldehyde microbeads were available on the company website with these features and, overall, with the range size 1-5 µm. The suitability of this MP for our purpose was certified by the use of the same fluorescent microbeads in previous studies on the same topic (Cattaneo et al. 2023, 2024; Zarantoniello et al. 2024).
- Discussion Section: Add information regarding the feed conversion ratio.
Authors’ reply: information regarding the feed conversion ratio have been added
- Use of abbreviation: If abbreviations are defined once, do not repeat their definitions. For example, in line 213, specific growth rate (SGR), relative growth rate (RGR), and feed conversion ratio (FCR) are defined. From line 269 onwards, use SGR, RGR, and FCR without redefining them.
Authors’ reply: Corrected